# Green Awareness in Action—How Energy Conservation Action Forces on Environmental Knowledge, Values and Behaviour in Adolescents' School Life

**Michaela Maurer [1],\*, Pavlos Koulouris [2] and Franz X. Bogner [1]**

[1] Didactics of Biology, Z-MNU (Centre of Math & Science Education), University of Bayreuth, NW-1, Campus, D-95447 Bayreuth, Germany; franz.bogner@uni-bayreuth.de

[2] Research and Development Department, Ellinogermaniki Agogi, GR-15351 Athens/Pallini, Greece, pkoulouris@ea.gr

\* Correspondence: Michaela.Maurer@uni-bayreuth.de

**Abstract:** Affordable, reliable, sustainable and modern energy consumption is a crucial goal of the Agenda 2030. To raise each citizen's awareness for more effective energy consumptions, proper education is necessary. The classroom project GAIA (Green Awareness in Action) was designed to change energy consumption patterns to pursue green behaviour. The class-wise aim was to improve schools' $CO_2$-balance and to promote environmentally sustainable behaviour without impacting school life quality. Our target group were sixth graders ($N = 132$, $M = 11.03$, $SD \pm 0.23$, 53.4% = girls) of one Greek school. To monitor the project's effect, a pre- and post-test design was applied to measure environmental literacy regarding environmental knowledge, attitudes/values and behaviour. A regression analysis revealed that students with poor previous knowledge reached higher learning effects compared to those with good previous knowledge. Related to the environmental knowledge types, an ANCOVA analysis revealed a knowledge gain in action-related and effectiveness knowledge. The overall learning effect correlates positively with pro-environmental preference (high scores in preservation, low scores in utilisation) and negatively with weak pro-environmental preferences. Anthropocentric (utilitarian) preferences primarily focussing on nature exploitation have considerably decreased. The project illustrates how far individual behaviour can be targeted in green educational initiatives.

**Keywords:** Environmental Education (EE); Education for Sustainable Development (ESD); environmental knowledge types and values; Environmental Literacy (EL); moderated regression; sustainability

## 1. Introduction

### 1.1. Green Educational Initiatives

Conferences in Stockholm [1] and Rio de Janeiro [2] strongly recommended that conservation efforts should integrate cognitive, affective and psychometric aspects in formal and informal settings [3]. Developing Agenda 21, Environmental Education (EE) was formally turned into Education for Sustainable Development (ESD) retaining its initial aim to support environmental protection. ESD comprises three components: environmental, economic and social sustainability (three-pillar model of sustainability) [4], raising awareness for worldwide sustainability with respect to present and future generations [5,6]. In 2015, the United Nations updated the Agenda 2030's expectations, including peace and international cooperation with all nations [7]. The Foundation for Environmental

Education (FEE), a non-governmental, non-profit organisation founded in 1981, promotes educational programmes for young people who approach an environmentally sustainable lifestyle. Its programmes comprise Blue Flag, Eco-Schools, Learning about Forests (LEAF), Young Reporters for the Environment and Green Key, although the latter does not include in-class teaching. Eco-schools are the most popular among all programs, acting at a global level with an action-based learning plan. Today, over 59,000 schools in 68 countries are taking part [8]. Current topics are water, waste avoidance, saving energy, biodiversity, transport as well as sustainable mobility, health, noise and climate change.

Green education initiatives are also offered outside FEE for conventional schools in formal and informal education contexts for half a day or even several days long settings. Some schools additionally work together with organizations. This raises the question of the extent to which green education initiatives actually contribute to environmental awareness regarding EE or ESD. Thus, there should be room for discussing such green education initiative' applications. A valid and reliable psychometric measuring instrument could point out some benefits.

Various conferences and agreements of global and historical relevance [1,2] have the common goal to raise citizens' awareness for a more conscious handling of problems associated with environmental and natural resources. Environmental awareness comprises three components: environmental knowledge, values and behaviour. Usually, the examination of knowledge, attitude, behaviour components and other variables are compared using psychometric measuring instruments to assess if they are influenced by such green educational initiatives. When applying bioenergy-modules [9], for example, 10th graders revealed a knowledge gain after a half-day long intervention. That gain, in turn, depended on (reported) behaviour and desirable attitude-set preferences (preservation and utilisation). Knowledge gains for girls were positively correlated with preservation and (reported) behaviour scores, whereas for boys there were only positive correlations with low utilisation's scores. Another intervention using a climate change module has shown significant short-term and long-term knowledge gain for 10th graders, while connectedness with nature and environmental attitude-sets played a substantial role for that gain [10,11]. A one-week environmental education outreach program on water issues reported similar changes regarding environmental knowledge and values for fourth and sixth graders [12].

GAIA (Green Awareness in Action) was a related green education initiative realised as part of a European research project on achieving behavioural changes for sustainability and energy awareness in schools. At one particular school in Greece, GAIA initiatives promoted problem-based learning, case-based teaching and discovery learning to increase sustainable behaviour of sixth-graders. Our analysis of this intervention included the sub-scale 'energy' which enabled us to detect changes in behaviour and to apply behaviour as a moderator of other variables (e.g., values). A first pilot testing [13] revealed close links between environmental behaviour and values. For environmental behaviour and environmental knowledge, there were only weak correlations. Taking cooperative action to change schools' energy consumption patterns was the key role of the intervention explained in the following.

### 1.2. Green Awareness in Action (GAIA) Intervention

The GAIA project, which was carried out by a multidisciplinary consortium consisting of nine partners from five European countries, involved in-class activities to foster sustainability awareness and energy saving. The major aim was to help students understand the impact of individual energy consumption and to promote changes in their habits. Deploying an Internet of Things (IoT) infrastructure in the participating schools, the project gathered data of energy consumption and sustainable infrastructure of classrooms. The GAIA Internet of Things (IoT) platform combines sensing, web-based tools and gamification elements, allowing the educational community to work with data produced by school buildings (e.g., temperature, relative humidity, illuminance, motion detection, noise level, electrical power consumption). The IoT installation in each school consists of a multitude of IoT nodes (sensors and meters) communicating with cloud-based services via gateway devices [14]. Access to this data, combined with tailor-made educational methods (e.g., computer-

based learning, ta 1a), tools and materials enabled school communities to monitor energy efficiency results.

The overall learning aims of GAIA, using the project's developed infrastructure and methodologies, were: (i) to raise awareness regarding the need to save energy and possibilities to implement energy-saving methods in everyday school life (students, educators, other staff) and wider communities (families, local communities); and (ii) to encourage environmentally-friendly behaviour, which can contribute to increased energy efficiency. The project was designed and implemented as an educational initiative beyond merely informing about energy efficiency, and involved students, teachers, and building managers in monitoring and reducing energy consumption in schools. Thereby, all decisions based on their previously set goals were not forcefully implemented, but carried out voluntarily. Everyone was encouraged to experiment with and adopt behaviours that proved effective.

The GAIA project was considered to stimulate three actions. First, to understand and monitor their own and others' behaviours affecting energy consumption in their school building; second, based on this, to make informed decisions and take action to increase energy efficiency; and third, to observe and analyse the impact of their actions on energy consumption and their effect on comfort, functionality and smooth functioning of school life.

GAIA initiatives were realized in 25 schools in Greece, Italy and Sweden. Our study focuses on the activities realized in the seven classes of the sixth grade (11-year-old students) of one of the participating Greek schools, in the greater area of Athens, led by the environmental education teacher.

Subjects were sixth graders (*N* = 132 students) who attended learning activities embedded in their regular environmental education course, for about 19 teaching hours over a period of 10 weeks in the first term of school year 2018–2019. Through observation, simple calculations and accessing as well as interpreting data provided by the GAIA infrastructure, they systematically monitored their use of electricity in the classroom. In consequence, students analysed how energy consumption changed throughout a fixed time slot and how it differed among classrooms (Figure 1b). After having discussed their data and shared decisions, they reconsidered their use of electric lightning (e.g., switching it off during daytime or when leaving the classroom). In addition, the building's orientation was taken into account to use daylight as effectively as possible. At the end of the week, students reduced their overall energy consumption by 50% and shared their success with the rest of the school community (Figure 1c).

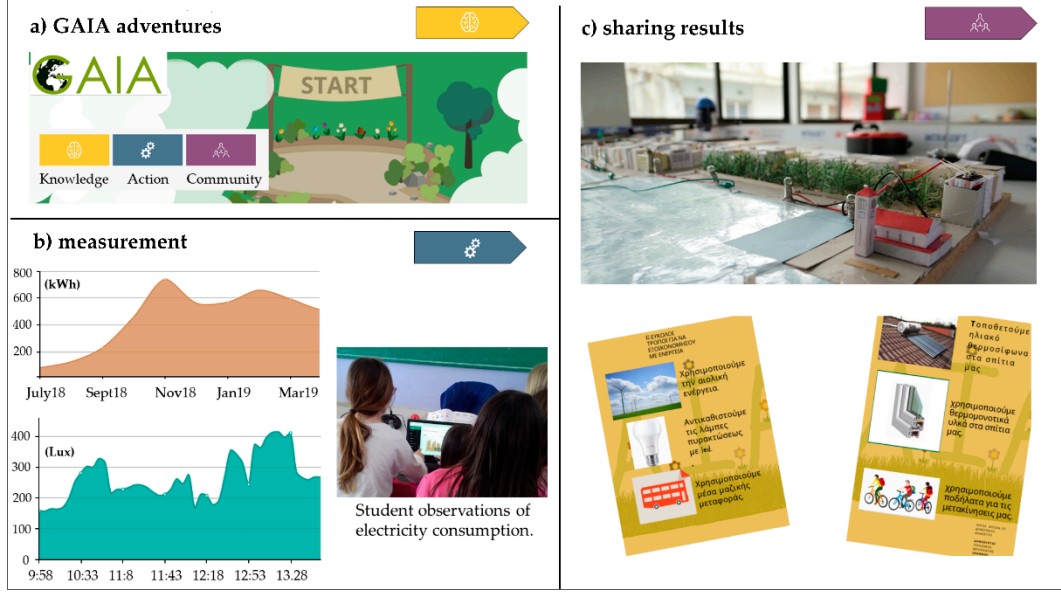

**Figure 1.** Green Awareness in Action (GAIA) progression: (a) knowledge generates awareness (e.g., through an independent tutorial program [15]), (b) action regarding observation of energy consumption and light intensity, including aberrations related to classroom conditions, and (c)

engagement via hands-on in-class activities and sharing ideas. Diagrams and pictures are offered by the GAIA project (P. Koulouris).

### 1.3. Environmental Literacy (EL) Cornerstones

A variety of definitions for environmental literacy (EL) exists [16]. The term literacy initially referred solely to the ability to read and write. However, EL was specifically developed for environmental education and comprises four cornerstones: knowledge (relationships to environmental behaviour), attitudes/values, behaviour and skills. According to Weinert [17], competencies are described as the sum of skills and abilities to cope with everyday situations. Extensive research was conducted to find suitable measuring instruments for competencies. Roczen and colleagues [18] defined a competency model applying three pillars of EL: (i) copying skills, referring to the knowledge structure model [19], (ii) values as attitude-sets, linked to the Two Major Environmental Value model (2-MEV) scale [20] and (iii) individual behaviour covered by the General Ecological Behaviour (GEB) scale [21]. This competency model with its three pillars was pilot-tested within the GAIA context [13], by accompanying appreciation (APR) [22] to the attitude-set. The following sections give a brief outline of the instruments we used.

(i) Since the 1990s, programme assessments have shown varying environmental knowledge gains (e.g., [23,24]). Various studies assumed that environmental knowledge alone is insufficient to measure an ecological lifestyle, as it does not automatically lead to action (e.g., [25]). Frick and colleagues [19] have already described different types of knowledge, namely system (SYS), action-related (ACT) and effectiveness knowledge (EFF); the first one is considered weak whereas the others may have a high impact [25,26]. Moreover, SYS and ACT together are supposed to form the basis of EFF [18,19]. Some researchers agree that different knowledge types can be influenced by pro-environmental behaviour (e.g., [26,27]), but not exclusively [28]. Using the three knowledge types, Thorn and Bogner [29] revealed a knowledge gain after six months, observing, for example, cognitive knowledge acquisition regarding nature conservation after tenth graders had visited an ecosystem forest. Similar results could be obtained six weeks after participation in an outreach drinking water module for seventh-graders [30]. Despite increasing numbers of green- and eco-school initiatives, environmental knowledge types are still not sufficiently examined (e.g., [31–33]). The GAIA intervention provides an opportunity to analyse relationships between knowledge types (as we revealed in an earlier pilot study [13]) and how much they are influenced by a pre- and post-test design.

(ii) Attitudes/values are an important part of environmental knowledge (e.g., [34]). For years, a suitable, reliable and valid instrument [35] to measure adolescents' attitudes has been lacking. Bogner and colleagues developed the Two Major Environmental Value model (2-MEV) [20,23] with two orthogonal factors: preservation (PRE) and utilisation (UTL). Over the last decades, the pilot instrument with more than 60 items has subsequently been cross-tested within bi-national studies (e.g., [36]) and further validated with other scales, such as the individual risk-taking preferences [37]. Finally, a 20 items version was agreed upon [38]. Moreover, what is the instrument was subsequently confirmed by independent groups at different times (e.g., [39–43]) accepting its ecocentric and anthropocentric views. The scale was frequently used to analyse the effectiveness of educational programs to provide recommendations for educational implementation efforts [44,45]. Based on green initiatives, Boeve-de Pauw and Van Petegem [46] reported that utilisation tendencies were lower in eco-school students than control school students. In consequence, values can also be a predictor of environmental knowledge as well as their willingness to learn (e.g., [47,48]). Relationships between other variables are expected [42]. Although further research will have to be conducted, green educational initiatives such as GAIA may support relationships between values, knowledge and environmental behaviour.

(iii) Various studies (e.g., [49,50]) describe that pro-environmental behaviours improve with growing environmental knowledge and attitudes. Other studies displayed changes, though not always significant, in environmental behaviour after educational interventions (e.g., [41]) or green educational initiatives (e.g., [46]). Nevertheless, one has to keep in mind that most cited examples

refer to different measuring scales and, thus, do not allow for the comparability. Demographic factors (age, sex and social status) or external factors (e.g., economic, social and cultural reasons), as well as internal factors (e.g., motivation, values and responsibilities and priorities), could be the underlying cause [51]. In the late 1990s, the General Ecological Behaviour (GEB) scale was established for adults [52] and subsequently adapted for adolescents in 2007 [21]. It comprises six pro-environmental sub-scales (consumerism, energy, mobility and transport, recycling, vicarious behaviour and waste avoidance) and was confirmed independently at different times (e.g., [53]). Using this instrument across national borders, we could observe various outcomes [50,54]. The GEB categories are all based on a possible relationship between attitudes and behaviour as an expression of motivation [55].

*1.4. Research Goals*

The objectives of our study were three-fold:

(i) to analyse the individual person estimate of each student, expressed in logits (natural logarithm of the ratio of correct to incorrect answer), for environmental knowledge and (reported) behaviour compared to differences between pre- (T0) and post-test (T1),

(ii) to observe how individual preferences (preservation and utilisation) interact with environmental knowledge/(reported) behaviour and differ after participation, and

(iii) to analyse the interaction of (reported) behaviour with environmental knowledge as a moderator.

## 2. Methods and Procedures

*2.1. Sample*

Our subjects were 132 Greek sixth-graders (seven school classes), in their last year of primary school ($M$ = 11.03, $SD$ ± 0.23, 53.4% = female), completing a paper and pencil questionnaires twice: first, after the beginning of the school year (pre-test, T0) and another time four months after intervention at the very end of the project (post-test, T1).

*2.2. Instruments*

Altogether three psychometric constructs were assessed:

(i) an individual environmental knowledge item set of 15 multiple choice question, adapted from the pilot study [13], including three different types (covariates) of system knowledge (SYS), action-related knowledge (ACT) and effectiveness knowledge (EFF) concerning the issues of "energy" (one correct answer for each multiple-choice question and five items of each knowledge type) (see Table 1),

(ii) 20 items regarding 2-MEV and APR scales [56], based on a five-point Likert scale ("1 = strongly disagree", "2 = disagree", "3 = partially agree", "4 = agree", "5 = totally agree") and

(iii) 21 items regarding the General Ecological Behaviour (GEB) scale containing four sub-categories: "energy", "mobility & transport", "recycling" and "vicarious behaviour" [21], based on a five-point Likert scale ("1 = never", "2 = seldom", "3 = sometimes", "4 = often", "5 = very often").

**Table 1.** Item examples for three different knowledge types: system (SYS), action-related (ACT) and effectiveness knowledge (EFF).

| Item Examples: |
| --- |
| SYS_6: Increasing demand for meat in a growing population reinforces the greenhouse effect. Which animals are responsible for the highest methane production? (a) fishes, (b) chicken, (c) pigs or (d) cows. |
| SYS_18: Which of the following electrical devices is the most energy-consuming one in the classroom? (a) projector, (b) computer + screen, (c) interactive whiteboard or (d) air conditioning. |

| |
|---|
| ACT_11: How can you save energy while cooking? (a) turn the stove to the lowest level, (b) use a lid on your pot, (c) turn the stove to the highest level or (d) order food using a delivery service. |
| ACT_21: How do you not save energy in the classroom? (a) do not use long curtains to prevent heat from escaping, (b) keep classroom doors closed, (c) close the windows in winter while heating the classroom or (d) during night time, turn the thermostat of the heating on its highest stage. |
| EFF_5: You urgently need new pants. How would you harm the environment most? (a) buy the article in a second-hand shop, (b) considering its origin and buy the article locally, (c) buy your article in an online shop choosing over-night express delivery or (d) going to the next store and buy new pants that please you. |
| EFF_15: How should buildings be painted in hot sunny areas like Italy to save energy? The buildings should be… (a) …lightly coloured to avoid overheating in summer, (b) …light or dark colours do not have any effect, (c) …dark colours to prevent overheating in winter or (d) …dark colours absorb the sun's heat. |

### 2.3. Statistical Analysis

Statistical tests were conducted using R (The R Foundation for Statistical Computing for Windows; Version 3.6.0 for Windows; www.r-project.org). The probabilistic Rasch model describes the likelihood to solve items via item difficulty and person ability. To estimate each person's attitudinal level, a dichotomous Rasch model for environmental knowledge and (reported) behaviour was developed, expressing results in logits (package eRM; for method, see [57]). Logits represent the ratio's natural logarithm between correct and incorrect answers. If logits were positive, the ratio shifted towards correct answers and vice versa. For this purpose, all polytomous items were converted into a dichotomous format. Response pattern followed a 5-point Likert scale (see instruments) ranging from 'strongly disagree (1)', 'disagree (2)' and 'partially agree (3)'. To represent unreliable pro-environmental preferences, we used 'zero' as the code. Analogous, 'agree (4)' and 'totally agree (5)', to represent pro-environmental preferences, we used 'one' as the code. For instance, we coded the item "I personally take care of plants" as one if the student ticked the item as 'totally agree'. For utilisation, we coded 'strongly disagree (1)' and 'disagree (2)' with 'one' to represent pro-environmental preferences. For example, we coded the item "Nature is always able to restore itself" as one if the student ticked the item as strongly disagree.

For 'partially agree (3)', 'agree (4)' and 'totally agree (5)', we coded unreliable preferences with 'zero'. All other items representing unreliable pro-environmental (reported) behaviour like 'never (1)', 'seldom (2)' and 'sometimes (3)' were also coded with 'zero'. Polytomous items such as 'often (4)' and 'very often (5)' which represent unreliable pro-environmental engagement were coded with 'one' (e.g., "I buy beverages in cans."). Reversed items were then used for coding (e.g., "In hotels, I have the towels changed daily.").

A simple regression analysis was applied to observe differences between person estimates in adolescents regarding environmental knowledge items. We also examined differences between girls and boys. Multivariate analyses of covariance's (ANCOVA) were used to determine group differences between covariates (packages effects). To test the homogeneity of variance between covariates, we applied Levene's test (packages car; for method, see [58]).

## 3. Results

### 3.1. Analyses of Quality and Environmental Knowledge

To analyse and visualise knowledge items (5 system- (SYS), 5 action-related (ACT) and 5 effectiveness knowledge (EFF)), we used a dichotomous Rasch model (eRm package, [57]). The likelihood ratio (LR) test confirmed the model's application (T0: LR = 9.045, $df = 14$, $p = 0.828$; T1: $= LR = 14.235$, $df = 14$, $p = 0.432$). The Wald test indicates one-dimensionality at item level without breaking the rules for T0's (T0$_{minimum}$ = −1.093, T0$_{maximum}$ = 1.29), whereas for T1 one item does. For other items, one-dimensionality is given (T1$_{minimum}$ = −1.82 T1$_{maximum}$ = 1.43). The weighted fit mean square (wMNSQ), which should range between 0.80 and 1.20 for multiple-choice tests [59,60], was acceptable

for pre- (T0) and post-test (T1): T0 $0.83 \leq$ wMNSQ $1.09 \leq$, T1 $0.79 \leq$ wMNSQ $\leq 1.17$. The item map illustrates that infit t-statistics applied to all items—except for one item in T0 (Figure 2A) and three items in T1—within the limits of ±1.95 against the latent dimension. According to Wang & Wilson [61], an item is only misfit, if wMNSQ and the infit t-statistic did not fit both. The joint Item Charateristic Curve-Plot (ICC plot) displayed a broad range of probability for environmental knowledge items as was plotted against the latent dimension (Figure 2B). The items' discriminatory power was comparable for each item, although it slightly shifted relative to item difficulties.

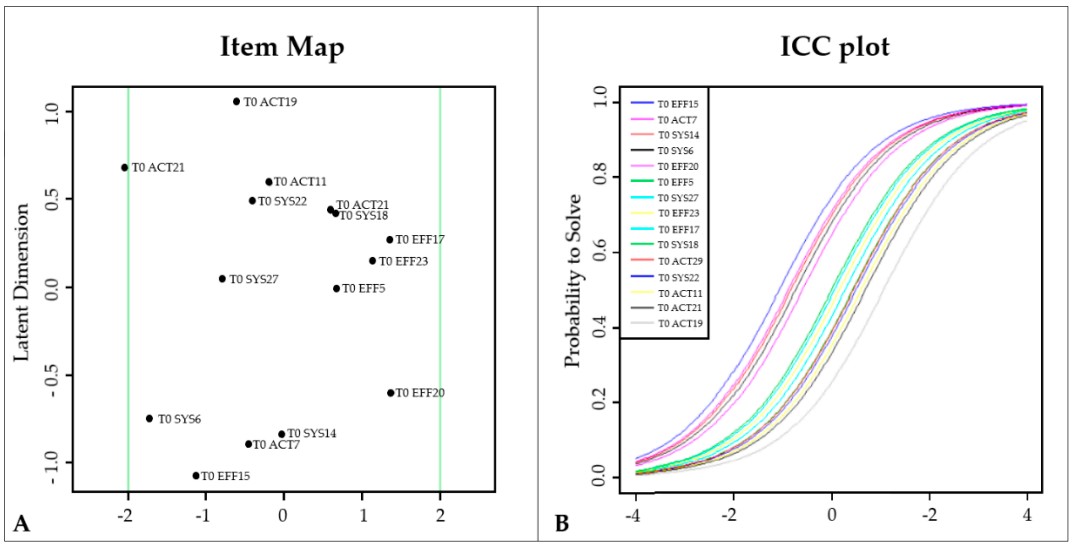

**Figure 2.** Visualising the infit t-statistic for knowledge items of the pre-test (**A**) and their characteristic item curves (**B**).

### 3.2. Person Estimate Promoting Environmental Knowledge

The person estimate indicates person performance and item difficulty based on pre- (T0) and post-tests (T1). To visualise differences, we used a simple regression analysis to observe variances of person estimate scores (logits) ($F = 12.66$, $df = 130$, $r^2 = 0.09$, $p = 0.001$) (Figure 3A).

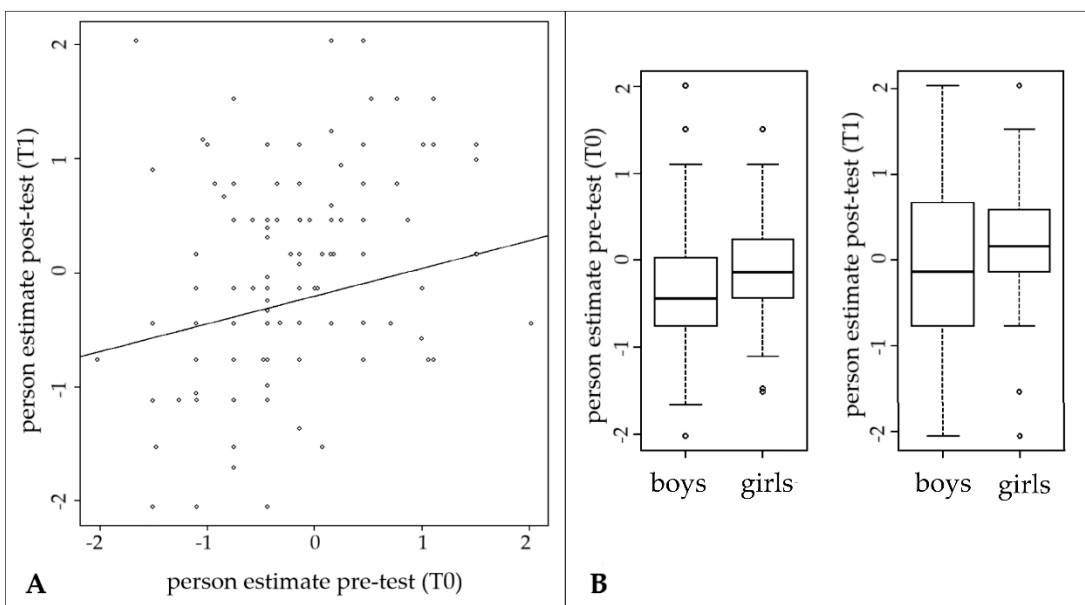

**Figure 3.** Environmental knowledge regression line in relation to participants' personal estimate ($N = 132$) (**A**) and correctly answered environmental knowledge items (**B**) for person estimate of boys and girls with regard to pre- and post-tests.

All participants were divided into two groups with respect to their logit scores to measure a possible increase in knowledge. Those with lower logit scores (group one, *N* = 66) answered about five items in the pre-test correctly and about seven in the post-test. Groups with higher logit scores (*N* = 66) solved approximately eight items in T0 and T1. Comparison between girls and boys showed that both sexes increased their knowledge after participation (Figure 3B), but that girls knew more than boys did.

To verify our assumptions, we applied multivariate tests of ANCOVA. For analyses of all three covariates (system (SYS), action-related (ACT) and effectiveness knowledge (EFF)), we used sum-scores (Figure 4A). Results are not based on a specific lesson; they refer to the 10 weeks after the intervention. While system knowledge decreased after having participated in the intervention, action-related and effectiveness knowledge increase. The test quality was acceptable (Figure 4B).

Levene's test indicates no homogeneity of variance between covariates (SYS$_{T0, T1}$: *df* = 126, *F* = 1.27, *p* = 0.28; ACT$_{T0, T1}$: *df* = 126, *F* = 0.82, *p* = 0.54 and EFF$_{T0, T1}$: *df* = 126, *F* = 0.21, *p* = 0.96). Classification of the person estimate as dependent factor for environmental knowledge did not reveal violations of homogeneity for regression slopes (SYS$_{T0, T1}$: *Sum Sq* = 0.49, *F* = 0.14, *p* = 0.94; ACT$_{T0, T1}$: *Sum Sq* = 3.50, *F* = 0.83, *p* = 0.48; and EFF$_{T0, T1}$: *Sum Sq* = 4.01, *F* = 0.96, *p* = 0.41).

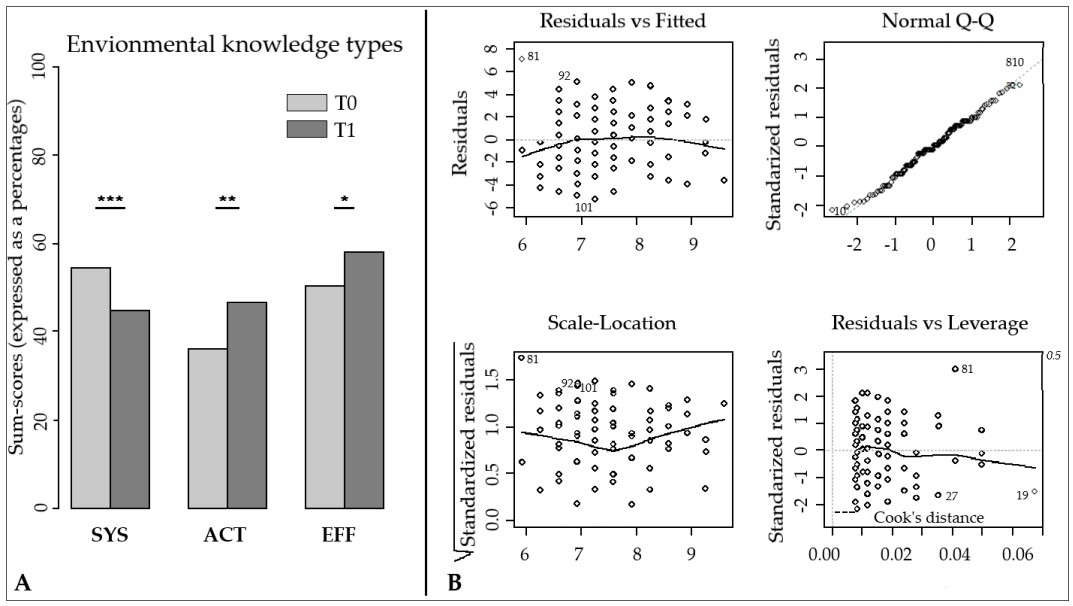

**Figure 4.** Correlation between the three knowledge types for pre- and post-tests (**A**) and test quality for analysis (**B**).

ANCOVA analysis (type III) showed that person estimate significantly affected covariates comparing single covariates of T0 and T1 with each other (SYS$_{T0, T1}$: *Sum Sq* = 16.02, *F* = 26.13, *p* > 0.001; ACT$_{T0, T1}$: *Sum Sq* = 21.92, *F* = 39.90, *p* > 0.001; and EFF$_{T0, T1}$: *Sum Sq* = 7.02, *F* = 9.01, *p* = 0.003) whereas for individual covariates two showed learning effects and one even a loss of knowledge (SYS$_{T0, T1}$: *r* = 0.30, *t* = 3.65, *df* = 130, *p* > 0.001; ACT$_{T0, T1}$: *r* = 0.24, t = 2.80, *df* = 130, *p* = 0.006 and EFF$_{T0, T1}$: *r* = 0.19, *t* = 2.24, *df* = 130, *p* = 0.03).

### 3.3. Nature Conservation Preferences and (Reported) Behaviour

In terms of attitudes, both groups (lower and higher values) displayed a significant knowledge gain (Table 2).

Classifying preservation as dependent factor for environmental knowledge indicates assumptions of homogeneity for regression slopes are not violated (SYS$_{T0, T1}$: *Sum Sq* = 1.79, *F* = 0.73, *p* = 0.48; ACT$_{T0, T1}$: *Sum Sq* = 3.00, *F* = 1.07, *p* = 0.35; and EFF$_{T0, T1}$: *Sum Sq* = 1.10, *F* = 0.42, *p* = 0.66). The same applies to appreciation (SYS$_{T0, T1}$: *Sum Sq* = 1.75, *F* = 0.15, *p* = 0.69; ACT$_{T0, T1}$: *Sum Sq* = 0.59, *F* =

0.51, *p* = 0.48; and EFF$_{T0, T1}$: *Sum Sq* = 1.40, *F* = 1.27, *p* = 0.26) and utilisation (SYS$_{T0, T1}$: *Sum Sq* = 0.77, *F* = 0.66, *p* = 0.42; ACT$_{T0, T1}$: *Sum Sq* = 0.001, *F* = 0.007, *p* = 0.98; and EFF$_{T0, T1}$: *Sum Sq* = 2.94, *F* = 2.80, *p* = 0.10).

ANCOVA analysis (type III) show that preservation did not significantly affect covariates (SYS$_{T0, T1}$: *Sum Sq* = 0.62, *F* = 0.54, *p* = 0.46; ACT$_{T0, T1}$: *Sum Sq* = 0.92, *F* = 0.78, *p* = 0.38; and EFF$_{T0, T1}$: *Sum Sq* = 0.33, *F* = 0.30, *p* = 0.58). The same applies to appreciation (SYS$_{T0, T1}$: *Sum Sq* = 1.75, *F* = 0.15, *p* = 0.70; ACT$_{T0, T1}$: *Sum Sq* = 0.59, *F* = 0.51, *p* = 0.48; and EFF$_{T0, T1}$: *Sum Sq* = 1.39, *F* = 1.27, *p* = 0.26) and utilisation (SYS$_{T0, T1}$: *Sum Sq* = 0.77, *F* = 0.66, *p* = 0.42; ACT$_{T0, T1}$: *Sum Sq* = 0.001, *F* = 0.007, *p* = 0.10; and EFF$_{T0, T1}$: *Sum Sq* = 2.94, *F* = 2.80, *p* = 0.10). We obtained comparable results for difference analyses of pre- and post-test mean values which distinguished between adolescents' lower and higher values (appreciation, preservation and utilisation) (Table 2).

**Table 2.** Environmental knowledge mean scores (*N* = 15 items) for pre- and post-test dependent on adolescents environmental values (lower and higher).

| Values | | T0 | T1 | Difference |
|---|---|---|---|---|
| Appreciation (APR) | Low | 6.85 | 7.73 | +0.88 |
| | High | 6.55 | 7.26 | +0.71 |
| Preservation (PRE) | Low | 6.50 | 7.30 | +0.80 |
| | High | 6.89 | 7.68 | +0.79 |
| Utilisation (UTL) | Low | 6.38 | 7.11 | +0.73 |
| | High | 7.02 | 7.88 | +0.86 |

*N* = 66 for each measurement.

Ecocentric tendencies remained at an almost equal level after having participated in our intervention for preservation (T0$_{preservation}$ = 62%, T1$_{preservation}$ = 63%), and slightly decreased for appreciation (T0$_{appreciation}$ = 37%, T1$_{appreciation}$ = 31%). Anthropocentric tendencies focused on exploiting nature did, however, not decrease (T0$_{utilisation}$ = 37%, T1$_{utilisation}$ = 47%, see methods).

Analysing item preferences of preservation regarding (reported) behaviour, participants with lower logit scores (group one) increased their environmental values after the intervention and were almost at the same level as participants with higher scores (group two) (Figure 5A). We observed similar effects for anthropocentric preferences (utilisation): Here, scores assessing tendencies to exploit nature decreased (Figure 5B). More items of 'disagree' and 'strongly disagree' were ticked in T1 than in T0 (see methods). No differences between pre-test and post-test were identified for appreciation preferences.

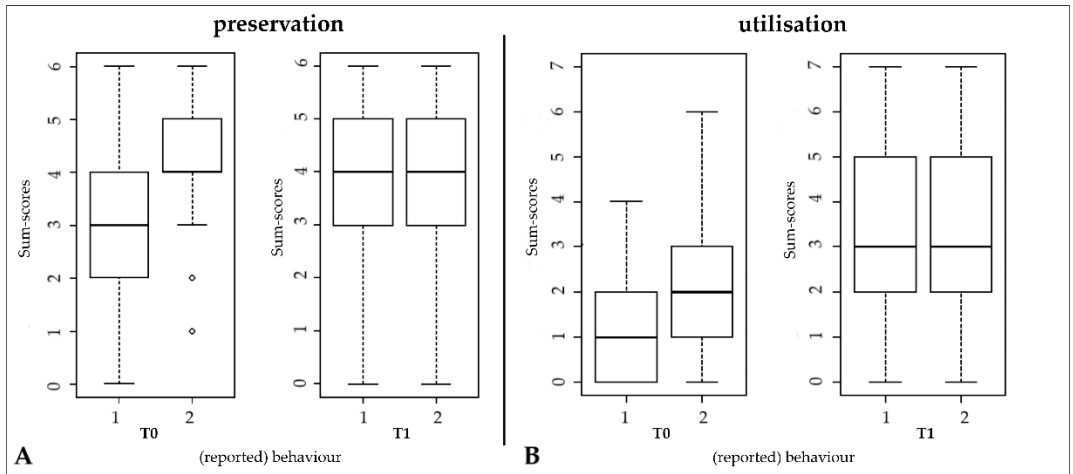

**Figure 5.** Ability to assess nature preferences (**A**) preservation and (**B**) utilisation via dependent logits displaying (reported) behaviour. 1 (x-axis) represents the group with the lower logit scores whereas 2 (x-axis) represents the group with the higher logit scores for both figures.

GEB as moderator shows an increased learning curve for environmental knowledge as its main interaction effect. Adolescents with lower logit scores have learned more compared to those with higher logit scores (Figure 6).

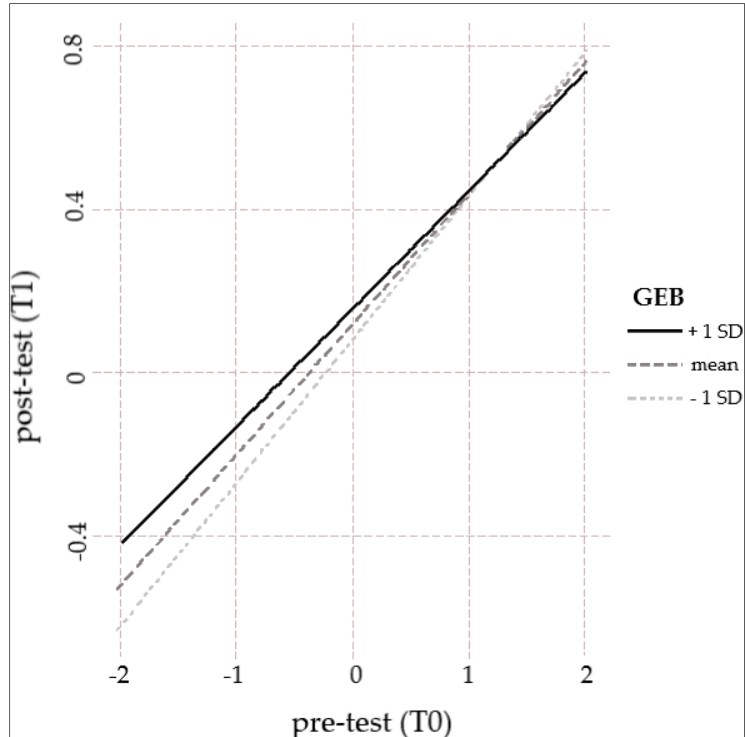

**Figure 6.** Conditional effect of the general ecological behaviour as moderator to solve environmental knowledge items between pre- and post-test.

## 4. Discussion

The major goal of our "green" module was affecting behaviour, promoting attitudes/values and fostering cognitive learning. Although this expectancy is in line with the literature [41,62], the GAIA (Green Awareness in Action) intervention did not meet all expectations: the overall environmental knowledge improved while values and behaviour did not. However, a detailed overview of the analyses could be of use.

### 4.1. Efficacy of Environmental Knowledge

Regarding the implementation of an environmental education program about water issues by Liefländer and colleagues [12], the highest increase in knowledge was observed for system knowledge and is much higher than in our study. Other researchers identified knowledge gains across all three knowledge types [33,41]. Poor basic pre-knowledge levels in system knowledge may provide a possible explanation for the discrepancy in our study [63,64]. After extensive analysis of energy consumption in the classroom, an increase in knowledge for action-related and effectiveness knowledge was the logical outcome of our study (Figure 4), although the actual increase was not particularly high. Action-related knowledge was the explicit goal of our intervention as it could potentially impact students' daily life. However, groups with lower logit scores (*N* = 66) increased their environmental knowledge scores whereas groups with higher logit scores (*N* = 66) did not. Of course, it would be desirable to specifically facilitate action-related knowledge as it could potentially impact students' daily life. In any case, it will be difficult to intrinsically motivate students to change their habits. Studies about adult consumers are already pointing out that most are unwilling to adjust their consumption patterns unless it is beneficial [65,66]. Grønhøj and Thøgersen [67] described motivation of young people as rooted in family descriptive norms. Generally, that some people are

more motivated to protect the environment (e.g., consuming and acting sustainably or saving energy) than others is explained by psychological factors of intrinsic motivation.

### 4.2. Environmental Values and Behaviour Preferences

Attitudes and informal education, although often critically discussed, do not impact pro-environmental behaviour [52]. In our case, we could show an interaction between pre- and post-test for environmental knowledge with (reported) behaviour as moderator (Figure 6). Poor motivation, thereby, seems to have an effect on learning as low-achievers (students who score below the baseline level were at a comparable level in the post-test) although they received lower environmental knowledge scores in the pre-test. This confirms our assumption, that action-related and effectiveness knowledge have a specific impact on low achievers. The respective attitudes/values seem to influence behavioural decisions [47,48]. Depending on higher and lower scores for preservation, appreciation and utilisation, it only affected cognitive achievement. In contrast to educational interventions in natural outreach setting [11,68], our in-class initiative focusing on energy consumption did not change attitudes toward nature. A possible explanation might be our target group's age: many participants were probably too young to understand the complex context of energy consumption patterns and $CO_2$ emissions and its impact on climate change. Boeve-de Pauw and Van Petegem [39] reported that environmental knowledge and utilisation preferences correlate negatively. Regarding environmental attitudes, their study displays an impact of social acceptability, not utilisation preferences, on preservation preferences. Our results suggest that tendencies to exploit nature (utilisation) significantly decreased throughout the intervention (Figure 5): utilizers may see an advantage in saving energy. Therefore, utilisation preferences had a larger effect than preservation or appreciation did. Our long-term intervention has influenced attitude but did not produce changes of attitudes which other interventions were able to achieve [24]. Eco-school projects, which mostly involve only a few classes, follow a similar pattern. Green educational initiatives will help to support raising awareness, but the decision to protect the environment should be an individual choice [67].

### 4.3. Limitation and Directions for Future Research

For reasons of comparability, we have limited the sample to one school (same teachers, same activities). We only included paper and pencil questionnaires filled in at both testing points despite considerably reducing the number of questionnaires through non-participants (reason of illness or non-participating). Rasch analysis was applied to all environmental knowledge items compared to behavioural items but was not carried out for each sub-scale. Unlike other interventions with three-fold testing cycles (pre-tests a few weeks before the intervention started, post-tests directly after the intervention and retention tests usually more than six weeks after the intervention), our testing schedule did not allow further insight into the knowledge acquisition. After participation in green initiatives, students have the largest knowledge gain which usually decreases shortly thereafter. In consequence, our monitoring procedure was limited to the beginning and end of the entire intervention phase. This is different from other studies and may indicate long time learning. Rasch analysis was applied to all environmental knowledge items versus behavioural items analyses but was not carried out for each sub-scale.

The GAIA project is an example of how to raise student's awareness of more sustainable energy consumption. Our results displayed that GAIA was a successful intervention. As it only entails minor technical effort and provides an application of materials [15,68], teachers in other schools can easily adapt its concept. Previous studies have already revealed [69] that interventions implemented in different classes or schools, but with similar school types and age groups, still produce different results. Differences within heterogeneous classes are quite usual. Furthermore, personal moderators (difficulties), situational moderators (social pressure) and subjective versus social norms (expectation) can impact the outcome [70]. Green initiatives regarding ESD do not specifically aim at changing awareness and behavioural preferences. Their focus is on shaping students' skills including other factors (knowledge, motivation, attitudes, values and take action) [71] to provide space for own decisions (e.g., self-efficacy according to Bandura's self-beliefs) [72].

## 5. Conclusions

Based on the three pillars of Education for Sustainable Development (ESD) (as outlined in the above), GAIA (Green Awareness in Action) specifically considered the economic pillar, as reducing energy consumption in schools was the main goal of the project. Participation encouraged school classes to compete with each other with regard to saving energy. Applied measuring devices confirmed that energy consumption patterns changed measurably throughout the intervention [68]. Unfortunately, our study was not able to connect the empirical data with the on-site measurement of hardware data in classrooms. Potential reasons for behavioural change may be two-fold: (i) it is related to the values of preservation and utilisation and (ii) it functions as a moderator for environmental knowledge. Furthermore, levels of action-related and effectiveness knowledge improved throughout the intervention. At the same time, behavioural decisions should not be the only factor taken into account. The social pillar, based on cooperation between students, teachers and stakeholders, was equally represented in the GAIA project. The impact of educational projects exceeds the boundaries of school (e.g., family, friends, open-source internet of Facebook, Instagram or Twitter) also included the ecological pillar, to protect the environment by changing energy consumption patterns and minimising $CO_2$ emissions.

Green education initiatives such as GAIA (or e.g., eco-schools) attempt to optimise energy consumption by various methods (e.g., improving students' knowledge levels). Potential expenses for in-class activities within the GAIA project mostly involve one-off payments (e.g., Internet of Things techs (IoT) like power meters, environmental sensors and weather stations)) and sensing hardware which are not expensive and applicable in other contexts. Moreover, we should keep in mind that the individual benefit of educational interventions highly differs within heterogeneous student communities. Projects like GAIA can only act as extrinsic operators. Still, it is possible that values and behaviour do not change despite extrinsic operators. Everyone must intrinsically motivate themselves. Creating individual learning tools and assessing their impact on competency could be subject to further research.

**Author Contributions:** M.M. initiated the first draft. All authors subsequently worked on the manuscript. All authors have read and agreed to the published version of the manuscript.

**Funding:** This project through by the "Qualitätsoffensive Lehrerbildung" (Grant: 01JA1901, https://www.qualitaetsoffensive-lehrerbildung.de/) and European project 'GAIA' (Green Awareness in Action: grant agreement No. 696029) as well as by the University of Bayreuth. Additionally, the German Research Foundation (DFG) and the University of Bayreuth in the funding program Open Access Publishing funded this publication.

**Acknowledgments:** The authors are very grateful to all students and teachers involved in this study for their time and engagement. We also thank Florian G. Kaiser for valuable statistics advice and Tamara Roth for proofreading.

**Conflicts of Interest:** The authors declare no conflict of interest.

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
