# Peer review of "Green Awareness in Action—How Energy Conservation Action Forces on Environmental Knowledge, Values and Behaviour in Adolescents’ School Life"

_sustainability, doi:10.3390/su12030955_

Round 1

Reviewer 1 Report

Thanks for this interesting contribution. It´s worth publishing, please consider revising the points I found for improving your paper.

Logits: this term is first defined in line 199, but it used before (abstract line 20 & in RQi l165). Maybe explain it in simpe words at first appearance and in abstract or replace it with a more general term.

l202-211 I´m not sure whether I understood this paragraph: the use of "unreliable" in a methods section of a statistical paper can be misleading. Pleas consider other wordings. Pleas also check 210-211 - I did not get the point with the reversed items (maybe as a consequence of the paragraph before?).

In the results the figures need a few changes:

Figure 2 & line 227: why did you give a figure of TO but not T1?

fig. 3 B: subtext of  figure and text above figure say lower and higher groups, in fig.3 B in the boxplot the sex is the variable - therefore it does not match.

Fig.4 B: Please explain these graphs also in the text.

Fig.5 please specify group 1 & 2 in the subtext (lower and higher logit scores), the reader has to infer from the text above.

291 ad fig.5 B scores decreased?? (1-->3 and 2-->3)?

266-267 please check syntax

286 37%-->47% is an increase (you write it did not decrease)

Ad discussion:

In heading 4.1 you introduce "efficacy" as a construct for the first time? Did you mean effectiveness (used in line 313)? Please check the language in this whole paragraph 4.1.

minor changes:

19 theory based scales

140 leave out: ", what is more," - germanism ;-)

164 objectiveS

168 ii to observe how individual...

170 ... knowledge has A moderator.

193 please add: correct answers are written in italics

252 test indicateS

331: suggestion: knowledge have a specific impact on low achievers.

341 tendencies TO exploit

342 utilizerS

344 attitude but did not produce changes of attitudes

346 sentence: consider revision - what do you mean?

Author Response

Dear Reviewer 1

Thank you for your support which helps to improve the quality of this manuscript.

Please find our responses attached.

Kind regards.

Reviewer 2 Report

Overall, I think this is a solid contribution to the literature, mainly requiring work to the presentation/writing. The two main areas for improvement with respect to writing would be:

(1) Editing at the sentence level to improve English, remove superfluous words and expressions, and improve flow;

(2) Editing at the level of the paragraph, particularly through focusing on providing topic sentences and ensuring that the paragraph in question has a main focus and that focus is clearly stated. Relatedly, synthesis of evidence statements are needed where links to the research literature are presented, particularly in the introduction section.

In more detail:

The Introduction section is somewhat confusing. This section should clearly lay out the background information pertinent to the problem that the paper is going to address. However, it is unclear from the first few paragraphs how the different pieces of information included relate to one another. Is the GAIA project under the auspices of FEE Eco Schools programme? If so, this should be clearly stated. If not, a more general statement about school programs, supported by the international research literature should be made.

The section  reporting on KAB (pg 2, Lns 48-65)should include more synthesis statements. Does the opening line on green education initiatives pertain to all green ed initiatives internationally or just those offered through FEE or Eco Schools? Some eco organisations work with schools over long timelines, so if made in reference to international green orgs, the statement is incorrect.

I suggest incorporating Marcinkowski and Reid's recent synthesis: "Reviews of research on the attitude–behavior relationship and their implications for future environmental education research" (published 2019, Environmental Education Research).

Small note: Can you provide more detail on the Internet of Things set-up in a footnote? I suspect that readers who are not familiar with how this works would appreciate knowing how it functions.

The Results section requires some editing work, for style/readability. Each section requires a statement of findings, rather than simply recounting the statistical model outputs. (This may mostly require re-arranging, rather than generation of new text.) This does not have to veer into "interpretation" but improves readability, especially as not all researchers will necessarily be familiar with your models. In other words, the onus is on you to present your findings, which you derive from your model outputs.

Discussion: Great opening summary.

Author Response

Dear reviewer 2

Thank you for your support which helps to improve the quality of our manuscript.

Please find our responses attached.

Kind regards.

Reviewer 3 Report

This paper aims to monitor the effects of classroom project (GAIA) on environmental knowledge, values and behaviour among sixth graders. The topic is interesting from the perspective of sustainability as education is needed to raise citizens’ awareness for more effective energy consumptions. However, some revisions are needed to improve the manuscript before it will be acceptable in Sustainability. More detailed comments and suggestions follow:

Abstract

- conclusions are missing from the abstract (results section could be shortened)

- it is not clear what ‘low achievers’ mean

Introduction

- 1.2. GAIA intervention, pp. 2–3: the context of the study is missing from this section – where did the intervention take place? Moreover, is the GAIA project international or in Greece only? Who designed the project and when? Who took care of the project in the classes – a teacher or somebody else? How many schools participated the project?

Materials and methods

- 2.1. Sample pp. 4–5: how these 7 classes were selected for the study? Were all 7 classes from one school/city? Did all students of these 7 classes respond to the questionnaires (was it obligatory or voluntary to participate in the research)? How old the participants were?

Results

- some parts of the results section are difficult to read with all statistical information. Is it possible to clarify the text?

Discussion

the main results of the study should be repeated more explicitly in each section of the discussion p. 10, line 343-345: “Our long-term intervention may have produced attitude did not produce attitudes changes…”. There is something wrong with this sentence, does not make sense. p. 10, lines 346-347: “In general, we would expect younger students to have more beneficial environmental than older students”. Is one word missing from this sentence – environmental what? p. 11, lines 350-51: “We only included paper and pencil questionnaires filled in at both testing points despite, considerably reducing the number of participants.” What does this mean?

Conclusions

- practical conclusions could be added: e.g. how could schools or environmental educators utilize the results of the study?

Author Response

Dear reviewer 3

Thank you for your support which helps to improve the quality of this manuscript.

Please find our responses attached.

Kind regards.
